# The Role of Ewes’ Udder Health on Echotexture and Blood Flow Changes during the Dry and Lactation Periods

**DOI:** 10.3390/ani12172230

**Published:** 2022-08-30

**Authors:** Aikaterini Ntemka, Ioannis Tsakmakidis, Constantin Boscos, Alexandros Theodoridis, Evangelos Kiossis

**Affiliations:** 1Clinic of Farm Animals, School of Veterinary Medicine, Faculty of Health Sciences, Aristotle University of Thessaloniki, 54627 Thessaloniki, Greece; 2Laboratory of Animal Production Economics, School of Veterinary Medicine, Faculty of Health Sciences, Aristotle University of Thessaloniki, 54124 Thessaloniki, Greece

**Keywords:** ewe, blood flow, color Doppler, echogenicity, B-mode, dry period, subclinical mastitis, healthy udder, mammary artery

## Abstract

**Simple Summary:**

During lactation, the mammary glands of ewes are prone to infections caused by numerous pathogens. Consequently, mastitis and other udder abnormalities frequently occur. Unfortunately, udder inflammation, whether clinical or subclinical, it poses a significant barrier to livestock profitability by reducing milk production, decreasing milk quality, and suppressing ewes’ reproductive performance. The cost of subclinical mastitis often is greater than that of clinical mastitis. Thus, such conditions must be treated during the dry period, which is important for mammary gland repair and recovery. Moreover, an early diagnosis is crucial for effective treatment. Ultrasonography is a useful tool that is employed in the detection of echotextural and hemodynamic changes. During recent decades, there has been an increasing scientific interest in the contributions of B-mode and Doppler to the determination of physiological and pathological conditions occurring in mammary glands, which cannot be detected during a physical examination. Therefore, the aim of the present study was to investigate the alterations in ewes’ udders’ echotexture and blood flow during the end of lactation, the stages of dry period, and the first days of the postpartum period. A further objective was to investigate the hemodynamic and echotextural differentiations between ewes having healthy udders and ewes having subclinical mastitis.

**Abstract:**

The objective of the current study was to investigate the echotextural and hemodynamic changes of ewes entering the dry period with or without subclinical mastitis. B-mode and color Doppler ultrasonography were applied to 12 Chios ewes (6 with healthy udders (group A) and 6 with subclinical mastitis (group B)) before the dry period, during the dry period (the involution phase, steady state, and transition phase), and postpartum. The color Doppler of the mammary arteries was used to evaluate them according to the pulsatility index (PI), resistive index (RI), end-diastolic velocity (EDV), time-averaged maximum velocity (TAMV), blood flow volume (BFV), and artery diameter (D). Udder parenchyma images, analyzed by Echovet v2.0, were used to evaluate the mean value (MV), standard deviation (SD), gradient mean value (GMV), gradient variance (GV), contrast (Con), entropy (Ent), gray value distribution (GVD), run length distribution (RunLD), and long run emphasis (LRunEm). In the involution phase, the PI was higher in group B compared to group A (*p* ≤ 0.05). The PI and RI were higher postpartum, whereas the EDV, TAVM, and D were higher in the transition phase (*p* ≤ 0.05). Neither the period nor the ewe group affected the MV, SD, GMV, GV, Con, and GVD values (*p* ≤ 0.05). In the steady state, the LRunEm was higher in group B, but postpartum, it was higher in group A (*p* ≤ 0.05). In conclusion, B-mode and Doppler can reveal differences (i) between healthy ewes and ewes with subclinical mastitis and (ii) among the different periods studied. Further research is needed on the blood flow and echotexture indices of the udders of ewes with unilateral subclinical mastitis.

## 1. Introduction

In small ruminants, udder abnormalities before or during the dry period have a negative effect on udder health and milk production after parturition. Mastitis is a significant financial issue in a herd as it detrimentally affects both lactogenesis and milk quality. It has been stated that subclinical mastitis is the most common cause of decreased milk production in sheep flocks [1]. Hence, an early diagnosis is required for effective treatment, but prevention is fundamental, as well [2,3]. Thus, the dry period is an important phase as it contributes to udder repair and recovery after infections that could occur during the previous lactation. This contribution is even more notable when intramammary antibiotics are suspended to ewes entering the dry period with subclinical mastitis. Further, antibiotics administration reduces the incidence of new udder infections during the dry period since distension of the cistern or blood flow changes during the involution stage increase the risk of an infection [4]. 

During recent decades, ultrasonography has been emerged as a non-invasive and accurate method that is employed in the detection of echotextural and hemodynamic changes of the mammary gland. Both B-mode and Doppler ultrasonography provide information in real-time about the structure and the physiological and pathological conditions of the udder, the teat cistern, and the teat canal [5]. Moreover, ultrasonographic examination reveals disorders that are not detected during a physical examination [6]. A further application of B-mode ultrasonography has been reported by a recent novel study. Using Artificial Intelligence, the echotexture images of an udder can predict the milk yield and production stage of a dairy cow [7]. 

Normally, mammary gland parenchyma is depicted as a homogenous granular echogenic structure, whereas non-homogenous regions correspond to mastitis [8,9]. Specifically, increased echogenicity is recorded in the case of udder inflammation or degeneration [5]. Previous studies have revealed large hyperechoic zones in udder parenchyma during ultrasonographic examination after an experimentally induced *Staphylococcus aureus* mastitis infection in goats [10]. Moreover, the udder parenchyma was characterized as inhomogeneous hyperechoic or granular, having scattered thin hyperechoic septa, after the ewes were intramammarily infected with different *Mycoplasma agalactiae* mutants [11]. On the other hand, udder echogenicity has also been reported to be normally reduced during lactation [12,13,14]. Additionally, it has been indicated that echogenicity progressively decreased as the dry period was evolving [15]. Concerning the ultrasonographic investigation of the mammary glands of dairy goats, no differences between goats with healthy udders and those with clinical or subclinical mastitis were observed [16].

In a normal udder, several anechoic structures represent the lactiferous ducts or the vessels. Doppler examination contributes to differentiating between them. Blood flow provides the mammary glands with the substrates necessary for lactogenesis, playing a key role in the functional activity of udders [17,18]. It is noteworthy to mention that as milk synthesis and substrates requirements increase at the onset of lactation, subsequently, blood flow also increases [19]. On the other hand, stress or fasting can decrease udder blood flow [20]. Previous studies have demonstrated that lactating cows have significantly higher blood flow velocities in their milk veins than dry cows [21]. Concerning blood flow volume (BFV), it has been recorded as being higher in the very first days after parturition [22], or it has not been altered [23]. No differences were found between cows having positive and negative California mastitis tests (CMT) regarding the pulsatility index (PI), resistive index (RI), end-diastolic velocity (EDV), and systolic peak velocity (SPV). However, the time-averaged maximum velocity (TAMV) was lower in CMT-positive cows [24]. Regarding goats’ udders, Doppler ultrasonography has been employed to investigate blood flow in their mammary veins and to correlate it to milk yield [25,26]. Additionally, in goats with clinical mastitis, the PI was found to be higher and the EDV was lower compared to healthy goats [16]. In fact, very few studies have investigated a ewe’s udder blood flow. Specifically, blood flow into the mammary gland and the diameter of the external pudendal artery progressively decreased during involution [15], whereas blood flow progressively increased at the last stage of pregnancy [27,28] or immediately after parturition [28].

Taking into account the above-mentioned research, the aim of the present study was to investigate the alterations in a ewe’s udder echotexture and blood flow at the end of lactation, in the stages of the dry period (lasting 105 ± 2 days) (the involution phase, steady state, and transition phase), and in the first days of the postpartum period. A further objective was to investigate the hemodynamic and echogenicity differentiations between ewes having healthy udders and ewes having subclinical mastitis.

## 2. Materials and Methods

### 2.1. Animals and Experimental Design

A survey was performed on 12 Chios breed ewes stabled at the Clinic of Farm Animals, School of Veterinary Medicine, Faculty of Health Sciences, Aristotle University of Thessaloniki. All ewes were in their second lactation, and their individual milk yield during the last day of milking was <300 mL. At the start of the drying-off procedure, all ewes were clinically examined, with special attention given to their mammary glands. The California Mastitis Test (CMT) was performed on each mammary gland of every ewe. Among the 12 ewes, in 6 of them, the CMT was negative for both mammary glands (no gelling/thickening was observed, with <400,000 cells/mL). In the other 6 ewes, CMT was positive for both mammary glands (either mild or heavy gelling/thickening was observed, with scores of ≥+1 and with >400,000 cells/mL). 

Afterwards, milk samples were collected once from each mammary gland of all animals for microbiological culture. The microbiological culture and the biochemical identification of the isolated strains was performed according to Kiossis et al. [29]. Coagulase-negative staphylococci (CNS) were isolated by the infected samples (*Staphylococcus epidermidis*, *Staphylococcus simulans,* and *Staphylococcus chromogenes*). According to the results of the above-mentioned examinations, the ewes were separated in two groups: 6 ewes having healthy udders and 6 ewes having subclinical bilateral mastitis. 

Udder drying-off of all the ewes took place abruptly after the intramammary antibiotic suspension of Nafpenzal^®^ (Intervet International B.V., Boxmeer, Netherlands/Intervet Hellas S.A., Athens, Greece) (300 mg procaine benzylpenicillin, 100 mg nafcillin, and 100 mg dihydrostreptomycin).

Clinical examination, CMT, and microbiological cultures were also performed after parturition in both of the mammary glands of each ewe. Milk samples were collected once from every ewe before they had any contact with their newborn lambs. In all ewes, the CMT was negative for both mammary glands (no gelling/thickening was observed, with <400,000 cells/mL). In addition, all ewes were found to be free of CNS.

### 2.2. Ultrasonographic Examination of Ewes’ Udders

Ultrasonographic examination was applied at the end of lactation, during the stages of the dry period (lasting 105 ± 2 days) (involution phase: 13 days, steady state: 70 days, and transition phase: 20–24 days), and in the first days of the postpartum period. In the last week of lactation, measurements were performed daily. During the dry period, measurements were performed weekly. In the last week of the dry period and the first week postpartum, measurements were performed daily. The first postpartum ultrasonographic examination was applied after parturition.

B-mode and color Doppler ultrasonographic examinations took place in the milking parlor with the ewes in a standing position. Hairs on the udders were clipped to facilitate the procedure and to obtain improved images. Gel was applied on the udders and the transducers were placed transcutaneously. B-mode and Doppler were applied to both mammary glands of the ewes.

B-mode examination was performed by the Esaote MyLab™One (Esaote SpA, Genova, Italy/Endomedical G.P., Athens, Greece) portable ultrasound, equipped with a convex probe of 3.5 MHz. The probe was placed on the caudal surface of each mammary gland, along its longitudinal axis in order to evaluate the parenchyma [15,27]. The images of udder parenchyma were processed by Echovet v2.0 software (Aristotle University of Thessaloniki, Thessaloniki, Greece) to evaluate the mean value (MV), standard deviation (SD), gradient mean value (GMV), gradient variance (GV), contrast (Con), entropy (Ent), gray value distribution (GVD), run length distribution (RunLD), and long run emphasis (LRunEm).

Color Doppler ultrasonography was performed using the same equipment on the mammary arteries of each mammary gland. To locate the mammary artery, the probe was positioned medially on the caudal region of the udder, near the udder’s insertion to the abdomen [15,27]. The images of the uniform spectral waveforms from three consecutive cardiac cycles each of the mammary arteries were processed by MyLab software (Esaote.Star.App) (Esaote SpA, Genova, Italy/Endomedical G.P., Athens, Greece) in order to evaluate the pulsatility index (PI), resistive index (RI), end-diastolic velocity (EDV), time-averaged maximum velocity (TAMV), blood flow volume (BFV), and artery diameter (D).

### 2.3. Statistical Analysis

Statistical analysis was performed via SPSS^®^ (version 24.0, provided by the Aristotle University of Thessaloniki, Thessaloniki, Greece). The normality and homoscedasticity of the data were examined. A paired *t*-test was applied to compare the mean values of the blood flow indices between the two mammary glands. A further *t*-test was applied to compare the mean values of the echotexture parameters between the two mammary glands. A linear mixed ANOVA model with repeated measures was applied with the Bonferroni correction to determine the effect of the period and udder health group, as well as their interaction, on udder echotexture and blood flow. The mammary gland corresponds to the random effect of the mixed statistical model. The number (n) of ultrasonographic examinations (B-mode and Doppler) was 35 for each mammary gland of every ewe, independently of the udder health group. Specifically, the number of examinations was: 7 in the end of lactation, 2 in the involution phase, 10 in the steady state, 9 in the transition phase, and 7 in the first days of the postpartum period.

A value of *p* ≤ 0.05 was considered statistically significant.

## 3. Results

The values of the B-mode and color Doppler indices of the ewes’ mammary glands were considered together, as no statistical difference was observed between the left and right mammary glands (*p* > 0.05). It is noteworthy that the lack of differences between the two mammary glands during the ultrasonographic examinations of small ruminants has already been recorded by previous researchers [15,26].

The interaction between the periods and the udder health groups was not statistically significant for the B-mode and Doppler variables (*p* > 0.05).

Concerning the B-mode ultrasonographic examination, neither the period nor the ewe group affected the MV, SD, GMV, GV, Con, or GVD parameters (*p* > 0.05). In the steady state, the LRunEm was higher in the ewes with subclinical mastitis, but in the first days of the postpartum period, it was higher in the ewes with healthy udders (*p* ≤ 0.05) (Figure 1). The Ent was higher in the steady state compared to first days of the postpartum period (*p* ≤ 0.05) (Figure 2). The RunLD values were almost stable until the steady state, whereas during the transition phase, they increased (*p* ≤ 0.05) (Figure 3).

In the involution phase, the PI was higher in the ewes with healthy udders compared to the ewes with subclinical mastitis (*p* ≤ 0.05). Differences in the PI between the two ewe groups were not observed in any other phase (*p* > 0.05) (Figure 4). 

With the exception of the PI, no differences between the ewe groups were observed during any phase concerning the blood flow parameters (*p* > 0.05) (Figure 5, Figure 6, Figure 7, Figure 8 and Figure 9). Furthermore, no differences in any of the hemodynamic indexes (*p* > 0.05) were observed between the end of lactation and the involution phase (Figure 4, Figure 5, Figure 6, Figure 7, Figure 8 and Figure 9). Comparing udder blood flow during the transition phase and the first days of the postpartum period, the PI and RI were lower in the first days of the postpartum period (Figure 4 and Figure 5), whereas the EDV, TAMV, D, and BFV were higher (*p* ≤ 0.05) (Figure 6, Figure 7, Figure 8 and Figure 9). Additionally, the EDV, TAMV, D, and BFV values were higher in the transition phase compared to the steady state (*p* ≤ 0.05) (Figure 6, Figure 7, Figure 8 and Figure 9). The RI values were lower in the transition phase compared to the steady state (*p* ≤ 0.05) (Figure 5).

## 4. Discussion

According to the results of the present study, no echotexture differences were observed between the ewe groups except for the LRunEm, which was higher in the ewes with subclinical mastitis. This finding is in accordance with the results of previous studies on ruminants. Specifically, increased echogenicity has been noticed in cases of udder inflammation in cows [5]. Concerning small ruminants, hyperechoic formations have been detected during ultrasonographic examinations of udders after an experimentally induced *Staphylococcus aureus* or *Mycoplasma agalactiae* mastitis infection [10,11]. However, Santos et al. [16] observed that the echogenicity did not differ between goats with healthy udder and goats with subclinical mastitis. The aforementioned researchers performed echotexture analysis by evaluating the mean numerical pixel values and pixel heterogeneity, which are first-order statistical parameters, whereas in the present study, second- and higher-order statistical variables were assessed.

The risk of infection is increased during the involution phase due to structural changes that have been noticed inside the udders [4]. Nevertheless, in this case, intramammary antibiotic suspension after udder drying-off was supposed to play a key role. The effectiveness of antibiotics infusion during the dry period has already been demonstrated [30]. Nevertheless, infections that were not cured in the dry period have also been recorded. The etiology may be both infectious (*Staphylococcus aureus*, *Escherichia coli*, and *Pseudomonas aeruginosa*) and non-infectious (parity, udder morphology, duration of dry period, limited control, detection, and treatment of infections in lactation) [31,32].

The present study indicated that the udder state did not affect the echotexture, apart from the RunLD, which was altered through the dry period, and the Ent, which was lower postpartum. These findings are partially in line with those of Barbagianni et al. [27], who observed that the grey-scale intensity values of mammary parenchyma were progressively increasing during the last month of gestation, whereas they were decreased postpartum. On the other hand, previous investigations of ewes’ udders during the dry period revealed that the echotexture was progressively decreased as the dry period evolved [15]. Normally, udder echotexture has been recorded reduced during lactation [12,13,14]. The decrease of the udder parenchyma echotextural indices values after parturition could be attributed to milk production. It could also be associated with the histological changes that occur inside mammary glands and with the decrease in the number of epithelial cells [33].

The present survey showed that in the involution phase, the PI was lower in ewes with subclinical mastitis compared to ewes with healthy udders. Consequently, there is evidence that the blood flow was higher in the cases of intramammary inflammation. In fact, infection causes the dilatation of arterioles, capillaries, and venules, resulting in hyperemia and increased blood flowing inside udders [34]. Moreover, this local blood flow increase is associated with the accumulation of defense cells which control infection. On the contrary, Santos et al. [16] claimed that there was not any difference in any hemodynamic index studied (PI, RI, EDV, and SPV) between goats with healthy udders and goats with subclinical mastitis. Moreover, Rişvanli et al. [24] indicated that only the TAMV among the examined parameters (PI, RI, EDV, TAMV, and SPV) was lower in cows with udder inflammation. 

According to our results, the RI, D, and BFV did not change while ewes were passing from the involution phase to the steady state, whereas the TAVM and EDV decreased. Similarly, Petridis et al. [15] noticed that the blood flow into mammary glands progressively decreased during involution. On the other hand, Petridis et al. [15] stated that the D was lower after parturition compared to the involution phase. In fact, udder vessels begin to develop at the end of gestation, as does the diameter of mammary arteries to support the milk yield that follows parturition. At the same time, the concentration of vasoconstrictor hormones progressively decreases [35] while the concentration of vasodilatory oxytocin increases [36]. The different results could be attributed to the fact that we measured the diameter of the mammary artery while Petridis et al. [15] measured the diameter of the external pudendal artery before it branches to the mammary artery.

Taking into consideration the ample bibliography on ruminant mastitis, the most common method for subclinical mastitis diagnosis is the microbiological culture of a milk sample. It is still the most accurate method used for the isolation of the bacteria that cause udder infections. Furthermore, the most suitable method for confirming udder inflammation is measuring the somatic cells (SCC) in a milk sample. For a more accurate diagnosis, Fthenakis and Jones [37] revealed that in the milk sample of a healthy ewe’s udder, macrophages were mainly identified, whereas in the milk sample of an ewe’s udder with subclinical mastitis, neutrophils or lymphocytes were identified. 

In recent decades, researchers have had a special focus on the creation of techniques that will improve the diagnosis of mastitis at an early stage. Although the majority of these methods are time-consuming and expensive, they are promising enough. Molecular methods (e.g., polymerase chain reaction (PCR) and real-time PCR) that identify the nucleic acids of the pathogens responsible for ruminant mastitis have been widely used for an accurate diagnosis [38]. Further, the estimation of the specific enzyme activity (e.g., lactate dehydrogenase, alkaline phosphatase, and glucosaminidase) in milk samples may contribute to the diagnosis of subclinical mastitis [39]. Additionally, the measurement of milk electrical conductivity during milking in each mammary gland through milk tubes with conductimetres is a technique that has already applied to ruminants [40]. Moreover, the use of proteomics in mastitis detection has been employed to clarify the pathogenesis of mastitis caused by various agents and to identify the protein biomarkers that could be detected for a mastitis diagnosis [41]. Modern techniques, such as infrared thermography, have also been used in the detection of subclinical mastitis in cows, with controversial results [42].

The present study suggests the use of ultrasonography for the diagnosis of subclinical mastitis. Among the B-mode parameters studied, the LRunEm was higher in the udders of ewes with subclinical mastitis, confirming that the echogenicity is increased in cases of udder inflammation. Among the Doppler parameters studied, the PI was lower in the udders of ewes with subclinical mastitis, confirming that blood flow is higher in cases of intramammary inflammation. This study was a first step in the possible use of ultrasonography for the diagnosis of subclinical mastitis in ewes. Further investigation is needed for the application of ultrasonography in scanning the udders of ewes with clinical mastitis and for scanning the udders of ewes with mastitis (subclinical or clinical) in one of the two mammary glands. Such data, in combination with Artificial Intelligence, could assist microbiological examinations in automating the process for a rapid and early diagnosis, as well as provide valuable information on udder health status, milk yield, stage of lactation, etc.

## 5. Conclusions

In conclusion, B-mode and Doppler ultrasonographic examination could be an additional tool for the evaluation of (i) udder health during the dry period, as it can reveal differences in echotexture and blood flow between healthy ewes and ewes with subclinical mastitis, and (ii) changes in udder echotexture and blood flow among the different studied periods, independent of mammary gland health. Further research is needed on the blood flow and echotexture indices of the udders of ewes with unilateral subclinical mastitis.

## Figures and Tables

**Figure 1 animals-12-02230-f001:**
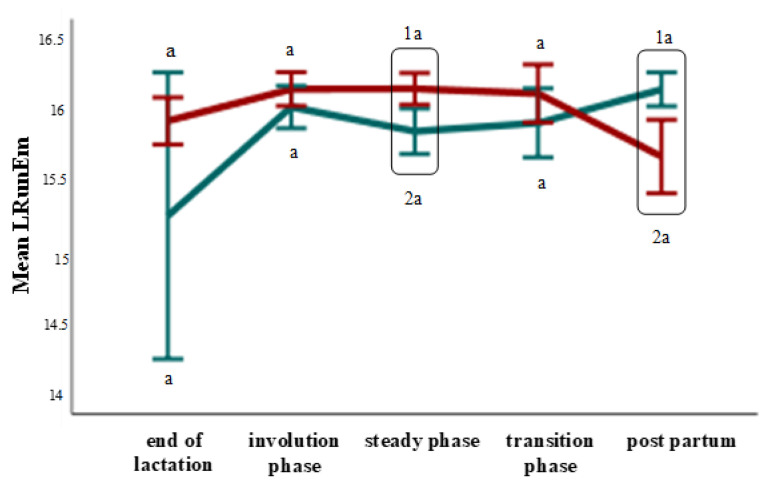
Long run emphasis changes among the periods and between the ewe groups (*p* ≤ 0.05). The different letter indicators (a, b) represent statistically significant differences between the periods, while the different number indicators (1, 2) represent statistically significant differences between the ewe groups in the same period. The green line represents the ewe group with healthy udders, while the red represents the ewe group with subclinical mastitis.

**Figure 2 animals-12-02230-f002:**
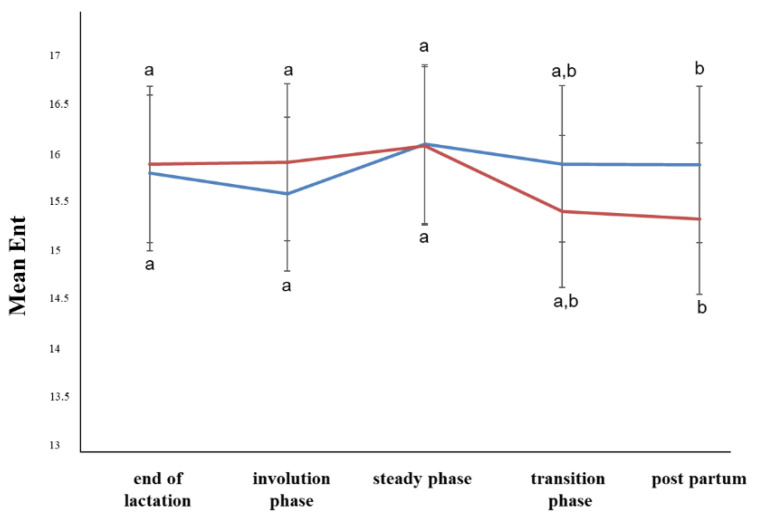
Entropy changes among the periods and between the ewe groups (*p* ≤ 0.05). The different letter indicators (a, b) represent statistically significant differences between the periods. The blue line represents the ewe group with healthy udders, while the red represents the ewe group with subclinical mastitis.

**Figure 3 animals-12-02230-f003:**
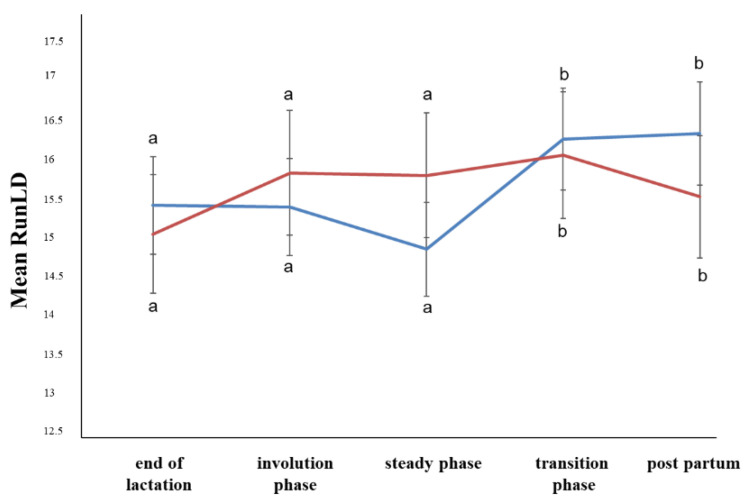
Run length distribution changes among the periods and between the ewe groups (*p* ≤ 0.05). The different letter indicators (a, b) represent statistically significant differences between the periods. The blue line represents the ewe group with healthy udders, while the red represents the ewe group with subclinical mastitis.

**Figure 4 animals-12-02230-f004:**
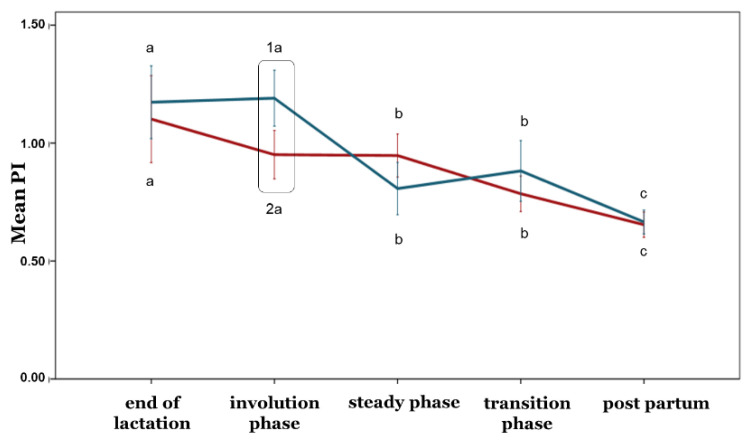
Pulsatility index changes among the periods and between the ewe groups (*p* ≤ 0.05). The different letter indicators (a, b, c) represent statistically significant differences between the periods, while the different number indicators (1, 2) represent statistically significant differences between the ewe groups in the same period. The green line represents the ewe group with healthy udders, while the red represents the ewe group with subclinical mastitis.

**Figure 5 animals-12-02230-f005:**
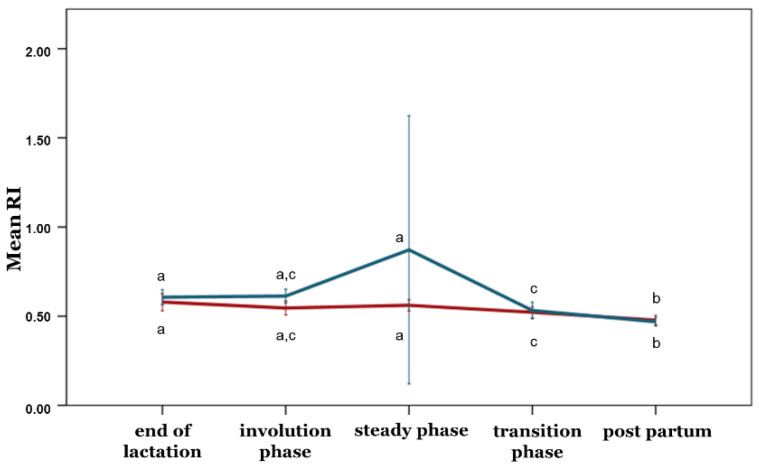
Resistive index changes among the periods and between the ewe groups (*p* ≤ 0.05). The different letter indicators (a, b, c) represent statistically significant differences between the periods. The green line represents the ewe group with healthy udders, while the red represents the ewe group with subclinical mastitis.

**Figure 6 animals-12-02230-f006:**
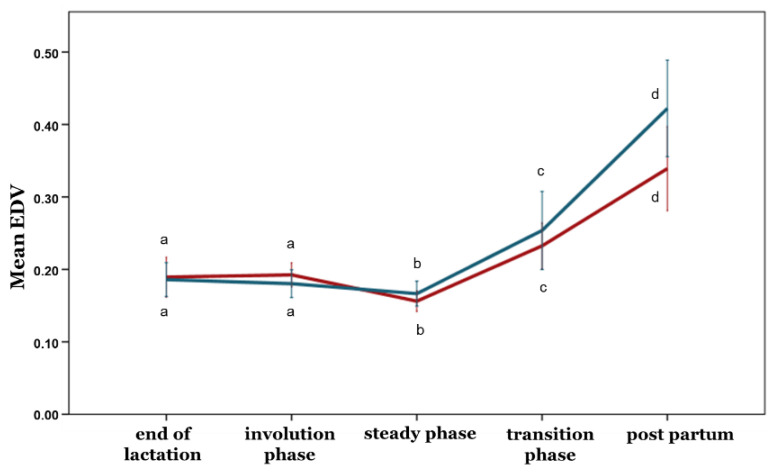
End-diastolic velocity changes among the periods and between the ewe groups (*p* ≤ 0.05). The different letter indicators (a, b, c, d) represent statistically significant differences between the periods. The green line represents the ewe group with healthy udders, while the red represents the ewe group with subclinical mastitis.

**Figure 7 animals-12-02230-f007:**
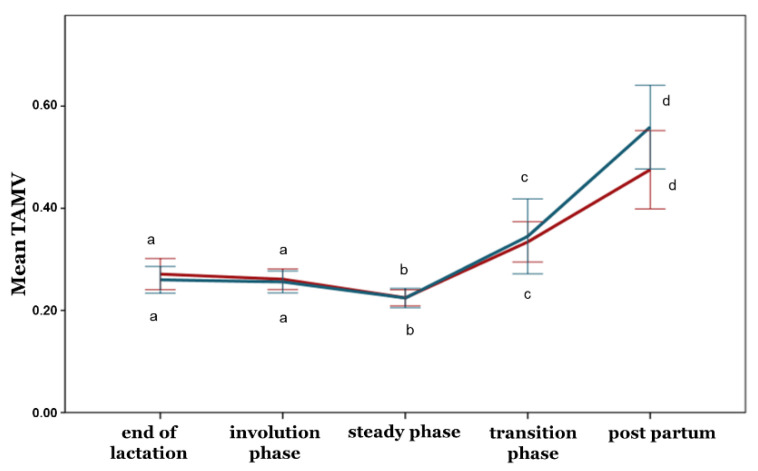
Time-averaged maximum velocity changes among the periods and between the ewe groups (*p* ≤ 0.05). The different letter indicators (a, b, c, d) represent statistically significant differences between the periods. The green line represents the ewe group with healthy udders, while the red represents the ewe group with subclinical mastitis.

**Figure 8 animals-12-02230-f008:**
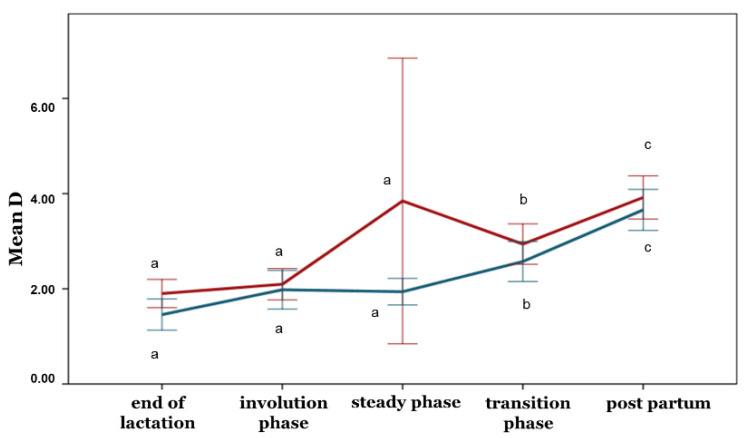
Mammary artery diameter changes among the periods and between the ewe groups (*p* ≤ 0.05). The different letter indicators (a, b, c) represent statistically significant differences between the periods. The green line represents the ewe group with healthy udders, while the red represents the ewe group with subclinical mastitis.

**Figure 9 animals-12-02230-f009:**
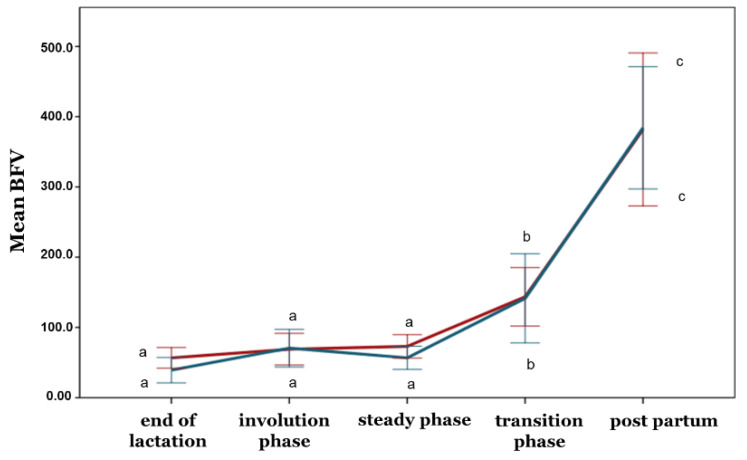
Blood flow volume changes among the periods and between the ewe groups (*p* ≤ 0.05). The different letter indicators (a, b, c) represent statistically significant differences between the periods. The green line represents the ewe group with healthy udders, while the red represents the ewe group with subclinical mastitis.

## Data Availability

The data presented in this study are available on request from the corresponding author. The data are not publicly available due to the project’s privacy restrictions.

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
