# Peer review of "The Role of Ewes’ Udder Health on Echotexture and Blood Flow Changes during the Dry and Lactation Periods"

_animals, 2022, doi:10.3390/ani12172230_

Round 1
Reviewer 1 Report
In general, this paper is clear, concise, and well-written though there are some errors grammatically. The only issue that I would like to ask is, which parameters can be further used for early diagnosis of subclinical mastitis thus making this procedure. Perhaps the author could suggest or summarize at the end for the readers.
Author Response
We appreciate and agree with reviewer's comments. Therefore, we responded to the reviewer's proposal. Please see the attachment.

Reviewer 2 Report
Dear authors,
This paper analyses the changes in ewes’ udder echotexture and blood flow in different periods of lactation to determine early the subclinical mastitis. The manuscript is well structured and written, the introduction provides sufficient background, and the research design is appropriate. However, several changes are necessary before publication. Concretely,
- The names of microorganisms should be written in italics (lines 72, 115, 238, 249, etc).
- The section “Materials and Methods” should include the information in subsections and expand the information on the techniques and animals used.
- Line 118: the authors should indicate the active principle of Nafpenzal and the concentration.
- Line 151: before carried out a parametric test, the authors must confirm the normality and homoscedasticity of the data.
- Section Discussion: the authors should extend discussion, including differences between the techniques used, among other.
- Figures: the authors indicate in the statistical analysis, the p-value<0.05 as significant. However, in all the figures, the value indicated is p-value ≤ 0.05. Please, unify it.
Author Response
We appreciate and agree with reviewer's comments. Therefore, we responded to all the reviewer's proposals step by step. Please see the attachment.

Reviewer 3 Report
See attached file

Author Response

(The authors gave the same response as above.)

Round 2
Reviewer 2 Report
Dear authors,
The manuscript has been corrected following the reviewer's comments and and it can be published in its current form,
Author Response
Thank you very much
Reviewer 3 Report
See file attached

Author Response
We appreciate and agree with reviewer’s comments. Therefore, we responded to all the reviewer's proposals, step by step.
